# Associations of semaglutide with incidence and recurrence of alcohol use disorder in real-world population

William Wang[1], Nora D. Volkow [2] ✉, Nathan A. Berger [1], Pamela B. Davis [3], David C. Kaelber [4] & Rong Xu [5] ✉

Alcohol use disorders are among the top causes of the global burden of disease, yet therapeutic interventions are limited. Reduced desire to drink in patients treated with semaglutide has raised interest regarding its potential therapeutic benefits for alcohol use disorders. In this retrospective cohort study of electronic health records of 83,825 patients with obesity, we show that semaglutide compared with other anti-obesity medications is associated with a 50%-56% lower risk for both the incidence and recurrence of alcohol use disorder for a 12-month follow-up period. Consistent reductions were seen for patients stratified by gender, age group, race and in patients with and without type 2 diabetes. Similar findings are replicated in the study population with 598,803 patients with type 2 diabetes. These findings provide evidence of the potential benefit of semaglutide in AUD in real-world populations and call for further randomized clinical trials.

An estimated 29.5 million or 10.6% of Americans ages 12 and older had an alcohol use disorder (AUD) in 2021[1]. AUD, which is responsible for more than 80,000 annual deaths in the USA is among the top 10 conditions associated with the largest global burden of disease[2]. Despite its large public health impact, there are only 3 medications for AUD approved by the FDA and 4 by the European Medicines Agency (EMA) and their therapeutic benefits are modest[3,4]. Thus, there is an urgent need to develop new medication for treating AUD.

Recent reports of reduced drinking in people being treated with glucagon-like peptide-1 receptor agonist (GLP-1RA) medications for T2DM or obesity have generated interest in the potential of these medications for treating AUD[5,6]. In particular semaglutide, a GLP-1RA approved for treating type 2 diabetes (T2DM) in 2017 and obesity in 2021, reduced drinking and relapse in alcohol-dependent rodents[7,8]. Anecdotal reports from patients prescribed semaglutide describe a reduced desire to drink[9] that have been subsequently corroborated by a report of reduced alcohol drinking with semaglutide and tirzepatide based on analyses of social media texts and follow up of selected

participants[10] and a case series reporting decreased symptoms of AUD in patients treated with semaglutide[11]. Moreover, a small clinical trial (*n* = 127) that evaluated the GLP-1RA agonist exenatide compared to placebo as an adjunct to standard cognitive-behavioral therapy, reported that exenatide significantly reduced heavy drinking days and total alcohol intake in a subgroup of patients with obesity[12]. However, as of now information on the clinical benefits of semaglutide for AUD prevention and treatment in real-world populations is still very limited. Here we took advantage of a large database of patient electronic health records (EHRs) to conduct a nationwide multicenter retrospective cohort study to assess the association of semaglutide with both the incidence and recurrence of AUD in individuals with obesity and with and without a prior history of AUD. We assessed the reproducibility of the findings in a separate cohort of patients with T2DM from non-overlapping time periods. We also compared patients who suffered from obesity who had T2DM (~33%) and those who did not (~67%); as well as patients with T2DM who suffered from obesity (~40%) and those who did not (~60%), to evaluate if there were potential

[1]Center for Science, Health, and Society, Case Western Reserve University School of Medicine, Cleveland, OH, USA. [2]National Institute on Drug Abuse, National Institutes of Health, Bethesda, MD, USA. [3]Center for Community Health Integration, Case Western Reserve University School of Medicine, Cleveland, OH, USA. [4]Center for Clinical Informatics Research and Education, The MetroHealth System, Cleveland, OH, USA. [5]Center for Artificial Intelligence in Drug Discovery, Case Western Reserve University School of Medicine, Cleveland, OH, USA. ✉e-mail: nvolkow@nida.nih.gov; rxx@case.edu

interactions on the effects of semaglutide in patients with these two co-morbid conditions. Outcomes were separately evaluated by age, sex, and race.

## Results

### Association of semaglutide with incident AUD diagnosis in patients with obesity and no prior history of AUD

The study population consisted of 83,825 patients with obesity who had no prior diagnosis of AUD and were for the first time prescribed semaglutide or non-GLP-1RA anti-obesity medications including naltrexone or topiramate in 6/2021–12/2022. The semaglutide cohort ($n = 45,797$) compared with the non-GLP-1RA anti-obesity medications cohort ($n = 38,028$) was older, had a higher prevalence of severe obesity and obesity-associated comorbidities including T2DM and lower prevalence of mental disorders, and tobacco use disorder. After propensity-score matching, the two cohorts (26,566 in each cohort, mean age 51.2 years, 65.9% women, 15.8% black, 66.6% white, 6.5% Hispanic) were balanced (Table 1). The semaglutide cohort ($n = 45,797$) compared with the naltrexone/topiramate cohort ($n = 16,676$) was older, had a higher prevalence of severe obesity and obesity-associated comorbidities including T2DM and a lower prevalence of mental disorders, and tobacco use disorder. After propensity-score matching, the two cohorts (15,097 in each cohort, mean age 49.2 years, 71.0% women, 17.2% black, 64.6% white, 6.9% Hispanic) were balanced.

Matched cohorts were followed for 12 months after the index event. Compared to non-GLP-1RA anti-obesity medications, semaglutide was associated with a significantly lower risk of recurrent AUD diagnosis (0.37% vs 0.73%; HR: 0.50, 95% CI: 0.39–0.63), consistent across gender, age group and race. Significant lower risks were observed in patients with T2DM and without T2DM (Fig. 1a). Compared to naltrexone or topiramate, semaglutide was associated with a significantly lower risk of incident AUD diagnosis (0.35% vs 0.78%; HR: 0.44, 95% CI: 0.32–0.61), consistent across gender, age group and race and in patients with and without T2DM (Fig. 1b).

### Association of semaglutide with recurrent AUD diagnosis in patients with obesity and a prior history of AUD

The study population consisted of 4254 patients with obesity who had a prior diagnosis of AUD and were for the first time prescribed semaglutide or non-GLP-1RA anti-obesity medications including naltrexone or topiramate in 6/2021–12/2022. The semaglutide cohort ($n = 1470$) compared with the non-GLP-1RA anti-obesity medications cohort ($n = 2784$) was older, included more women, had a higher prevalence of severe obesity and obesity-associated comorbidities including T2DM and lower prevalence of adverse socioeconomic determinants of health, mental disorders, and substance use disorders. After propensity-score matching, the two cohorts (1051 in each cohort, mean age 52.6 years, 41.5% women, 16.6% black, 66.2% white, 7.4% Hispanic) were balanced (Table 2). The semaglutide cohort ($n = 1470$) compared with the naltrexone/topiramate cohort ($n = 1430$) was older, included more women, had a higher prevalence of severe obesity and obesity-associated comorbidities including T2DM and lower prevalence of adverse socioeconomic determinants of health, problems with lifestyle, and substance use disorders. After propensity-score matching, the two cohorts (715 in each cohort, mean age 51.5 years, 40.7% women, 15.8% black, 67.7% white, 6.7% Hispanic) were balanced.

Matched cohorts were followed for 12 months after the index event. Compared to non-GLP-1RA anti-obesity medications, semaglutide was associated with a significantly lower risk of recurrent AUD diagnosis (22.6% vs 43.0%; HR: 0.44, 95% CI: 0.38–0.52), which was consistent across gender, age group and race. Significant lower risks were observed in patients with T2DM and without T2DM (Fig. 2a). Compared to naltrexone or topiramate, semaglutide was associated with a significantly lower risk of incident AUD diagnosis (21.5% vs 59.9%; HR: 0.25, 95% CI: 0.21–0.30), which was consistent across

gender, age group and race and in patients with and without T2DM (Fig. 2b).

### Association of semaglutide with incident and recurrent AUD diagnosis in patients with T2DM

The study population for the analysis of incident AUD diagnosis in patients with T2DM consisted of 598,803 patients with T2DM who had no prior diagnosis of AUD and were for the first time prescribed semaglutide or non-GLP-1RA anti-diabetes medications in 12/2017–5/2021. The semaglutide cohort ($n = 25,686$) compared with the non-GLP-1RA anti-obesity medications cohort ($n = 573,117$) was younger, had a higher prevalence of problems related to lifestyle, severe obesity, obesity-associated comorbidities and mental disorders. After propensity-score matching, the two cohorts (26,670 in each cohort, mean age 58.0 years, 45.3% women, 14.7% black, 60.3% white, 6.5% Hispanic) were balanced (Supplementary Table 1).

The study population for the analysis of recurrent AUD diagnosis in patients with T2DM consisted of 22,113 patients with T2DM who had a prior diagnosis of AUD and were for the first time prescribed semaglutide or non-GLP-1RA anti-diabetes medications in 12/2017–5/2021. The semaglutide cohort ($n = 668$) compared with the non-GLP-1RA anti-obesity medications cohort ($n = 21,445$) had a higher prevalence of adverse socioeconomic determinants of health, problems related to lifestyle, severe obesity, obesity-associated comorbidities and mental disorders. After propensity-score matching, the two cohorts (653 in each cohort, mean age 57.4 years, 25.9% women, 17.2% black, 55.5% white, 8.5% Hispanic) were balanced (Supplementary Table 2).

Matched cohorts were followed for 12 months after the index event. Compared to non-GLP-1RA anti-diabetes medications, semaglutide was associated with a significantly lower risk of incident AUD diagnosis (0.32% vs 0.52%; HR: 0.56, 95% CI: 0.43–0.74), consistent across gender, age group and race. Significant lower risks were observed in patients with and without a diagnosis of obesity (Fig. 3a). Semaglutide compared with non-GLP-1RA anti-diabetes medications was associated with a significantly lower risk of recurrent AUD diagnosis (23.4% vs 33.2%; HR: 0.61, 95% CI: 0.50–0.75), consistent across gender, age group, and race. Significant lower risks were observed in patients with and without a diagnosis of obesity (Fig. 3b). The significantly lower risk associations of semaglutide with both incident and recurrent AUD persisted, though slightly attenuated with overlapping confidence intervals, for the 2-year and 3-year follow-up (Fig. 3C).

## Discussion

Here we document a potential beneficial effect of semaglutide on both the incidence and recurrence of AUD in real-world populations. The findings were replicated in two separate populations with different characteristics, no-overlapping periods, and non-overlapping patients prescribed semaglutide: one with obesity and the other with T2DM. These beneficial effects are consistent with anecdotal reports that patients prescribed semaglutide describe reduced desire to drink alcohol while on the medication[9] and with recent clinical reports; one documenting reduced alcohol drinking with semaglutide or tirzepatide based on analyses of social media texts and follow up of selected participants[10], and another of decreased symptoms of AUD in a case series of patients treated with semaglutide[11]. It is also consistent with a small clinical trial study of the GLP-1RA drug exenatide, which significantly reduced heavy drinking days and total alcohol intake in patients with obesity[12] and with a register-based study in Demark showing that GLP-1RAs (though semaglutide was not included) compared with dipeptidyl peptidase 4 inhibitors (DPP4) were associated with lower incidence of alcohol-related events in 2009–2017[13]. It is also consistent with preclinical studies that documented reduced drinking in rodents exposed to semaglutide[8] and that prevented relapse in a rat model of alcohol dependence[7].

**Table 1 | Characteristics of the semaglutide cohort and the non-GLP-1RA anti-obesity medications cohort for the study population with obesity who had no prior history of AUD**

| | Before propensity-score matching | | | After propensity-score matching | | |
|---|---|---|---|---|---|---|
| | Semaglutide cohort | Non-GLP-1RA anti-obesity medications cohort | SMD | Semaglutide cohort | Non-GLP-1RA anti-obesity medications cohort | SMD |
| Total number | 45,797 | 38,028 | | 26,566 | 26,566 | |
| Age at index event (years, mean ± SD) | 53.2 ± 13.3 | 50.0 ± 15.2 | 0.22[a] | 51.2 ± 13.2 | 51.1 ± 15.1 | 0.004 |
| Sex (%) | | | | | | |
| Female | 62.3 | 66.0 | 0.08 | 65.9 | 65.8 | 0.001 |
| Male | 32.8 | 28.5 | 0.09 | 28.8 | 28.8 | 0.002 |
| Unknown | 4.9 | 5.5 | 0.03 | 5.3 | 5.4 | 0.007 |
| Ethnicity (%) | | | | | | |
| Hispanic/Latinx | 7.3 | 6.2 | 0.04 | 6.4 | 6.5 | 0.005 |
| Not Hispanic/Latinx | 69.4 | 74.0 | 0.10[a] | 71.8 | 72.0 | 0.003 |
| Unknown | 23.4 | 19.8 | 0.09 | 21.8 | 21.5 | 0.007 |
| Race (%) | | | | | | |
| Asian | 2.4 | 1.0 | 0.11[a] | 1.3 | 1.3 | 0.005 |
| Black | 16.2 | 15.7 | 0.01 | 15.8 | 15.8 | 0.001 |
| White | 64.9 | 67.1 | 0.05 | 66.7 | 66.5 | 0.003 |
| Unknown | 11.8 | 12.2 | 0.01 | 12.1 | 12.2 | 0.003 |
| Marital status (%) | | | | | | |
| Never Married | 12.1 | 16.6 | 0.13[a] | 13.9 | 13.8 | 0.001 |
| Divorced | 5.6 | 6.3 | 0.03 | 5.7 | 5.8 | 0.003 |
| Widowed | 3.0 | 3.3 | 0.02 | 3.1 | 3.1 | <0.001 |
| Adverse socioeconomic determinants of health (%) | 4.7 | 6.1 | 0.07 | 5.3 | 5.4 | 0.004 |
| Problems related to lifestyle (%) | 8.3 | 10.8 | 0.08 | 9.4 | 9.4 | <0.001 |
| Obesity categories (%) | | | | | | |
| Morbid (severe) obesity due to excess calories | 60.1 | 50.0 | 0.20[a] | 55.7 | 55.6 | 0.003 |
| Obesity, unspecified | 63.7 | 60.9 | 0.07 | 62.1 | 62.0 | 0.001 |
| Other obesity due to excess calories | 15.4 | 12.2 | 0.09 | 14.1 | 13.9 | 0.006 |
| BMI 30.0–30.9 | 6.2 | 7.7 | 0.06 | 6.8 | 6.8 | 0.001 |
| BMI 31.0–31.9 | 6.9 | 8.0 | 0.04 | 7.3 | 7.2 | 0.002 |
| BMI 32.0–32.9 | 7.7 | 8.7 | 0.04 | 8.0 | 7.9 | 0.001 |
| BMI 33.0–33.9 | 8.3 | 8.7 | 0.01 | 8.3 | 8.5 | 0.007 |
| BMI 34.0–34.9 | 9.0 | 8.6 | 0.01 | 8.7 | 8.7 | 0.001 |
| BMI 35.0–35.9 | 10.6 | 9.7 | 0.03 | 10.0 | 9.9 | 0.005 |
| BMI 36.0–36.9 | 9.6 | 8.6 | 0.03 | 9.1 | 9.0 | 0.002 |
| BMI 37.0–37.9 | 9.4 | 8.2 | 0.04 | 8.8 | 8.5 | 0.009 |
| BMI 38.0–38.9 | 9.1 | 7.8 | 0.05 | 8.6 | 8.4 | 0.006 |
| BMI 39.0–39.9 | 8.3 | 6.8 | 0.06 | 7.5 | 7.4 | 0.004 |
| BMI 40.0–44.9 | 22.9 | 20.4 | 0.06 | 21.9 | 21.8 | 0.003 |
| BMI 45.0–49.9 | 13.7 | 11.5 | 0.07 | 12.7 | 12.7 | 0.001 |
| BMI 50.0–59.9 | 9.5 | 7.9 | 0.06 | 8.7 | 8.7 | 0.001 |
| BMI 60.0–69.9 | 2.5 | 2.2 | 0.02 | 2.3 | 2.4 | 0.004 |
| BMI ≥70 | 0.9 | 0.9 | 0.003 | 0.9 | 0.9 | 0.002 |
| Family history of mental and behavioral disorders | 0.7 | 1.1 | 0.05 | 0.8 | 0.8 | 0.002 |
| Pre-existing medical conditions, procedures, medications (%) | | | | | | |
| Type 2 diabetes | 56.7 | 24.4 | 0.70[a] | 33.2 | 33.5 | 0.006 |
| Depression | 29.6 | 40.1 | 0.22[a] | 35.6 | 35.6 | 0.001 |
| Mood disorders | 34.9 | 49.0 | 0.29[a] | 42.6 | 42.4 | 0.003 |
| Anxiety disorders | 40.5 | 50.1 | 0.19[a] | 46.7 | 46.7 | 0.006 |
| Psychotic disorders | 1.2 | 2.3 | 0.08 | 1.7 | 1.7 | <.001 |
| Behavioral disorders | 9.4 | 8.9 | 0.02 | 9.7 | 9.4 | 0.009 |
| Disorders of adult personality and behavior | 1.2 | 1.9 | 0.06 | 1.5 | 1.5 | 0.001 |
| | 4.7 | 6.5 | 0.08 | 5.9 | 5.9 | <.001 |

**Table 1 (continued) | Characteristics of the semaglutide cohort and the non-GLP-1RA anti-obesity medications cohort for the study population with obesity who had no prior history of AUD**

| | Before propensity-score matching | | | After propensity-score matching | | |
|---|---|---|---|---|---|---|
| | Semaglutide cohort | Non-GLP-1RA anti-obesity medications cohort | SMD | Semaglutide cohort | Non-GLP-1RA anti-obesity medications cohort | SMD |
| Behavioral and emotional disorders with onset usually occurring in childhood and adolescence | | | | | | |
| Conduct disorders | 0.3 | 0.7 | 0.06 | 0.4 | 0.4 | 0.004 |
| Symptoms and signs involving emotional state | 5.0 | 6.5 | 0.07 | 5.7 | 5.6 | 0.001 |
| Chronic pain | 29.5 | 27.8 | 0.04 | 29.4 | 28.7 | 0.02 |
| Cancer | 36.5 | 28.7 | 0.17[a] | 32.8 | 32.4 | 0.009 |
| Cannabis use disorder | 1.2 | 2.5 | 0.09 | 1.7 | 1.6 | 0.004 |
| Opioid use disorder | 1.6 | 2.4 | 0.06 | 1.9 | 1.9 | 0.005 |
| Tobacco use disorder | 12.8 | 20.3 | 0.20[a] | 15.8 | 15.5 | 0.008 |
| Cocaine use disorder | 0.4 | 0.7 | 0.05 | 0.5 | 0.4 | 0.008 |
| Other stimulant disorders | 0.4 | 0.9 | 0.06 | 0.5 | 0.5 | 0.004 |
| Other psychoactive substance related disorders | 1.0 | 1.9 | 0.08 | 1.3 | 1.3 | 0.002 |
| Hypertension | 75.1 | 70.1 | 0.11[a] | 71.1 | 71.1 | <.001 |
| Disorders of lipoprotein metabolism and other lipidemias | 73.6 | 61.6 | 0.26[a] | 67.3 | 67.0 | 0.007 |
| Hyperlipidemia | 52.4 | 42.3 | 0.20[a] | 46.3 | 46.0 | 0.006 |
| Hypercholesterolemia | 22.8 | 16.1 | 0.17[a] | 19.1 | 18.6 | 0.01 |
| Ischemic heart diseases | 16.9 | 13.6 | 0.09 | 14.3 | 14.1 | 0.006 |
| Other forms of heart disease | 31.3 | 30.2 | 0.02 | 30.0 | 29.7 | 0.007 |
| Cerebral infarction | 3.1 | 3.4 | 0.02 | 3.2 | 3.0 | 0.009 |
| Cerebrovascular diseases | 7.0 | 7.5 | 0.02 | 7.0 | 6.9 | 0.001 |
| Substance abuse treatment | 0.0 | 0.1 | 0.03 | 0.0 | 0.0 | <.001 |
| Psychotherapy | 4.4 | 4.9 | 0.02 | 4.9 | 4.8 | 0.005 |
| Baclonfen | 3.9 | 4.3 | 0.02 | 4.3 | 4.2 | 0.002 |
| Gabapentin | 23.2 | 23.5 | 0.006 | 23.5 | 23.0 | 0.01 |
| Medical visit types (%) | | | | | | |
| Outpatient | 94.1 | 91.3 | 0.11[a] | 93.2 | 93.2 | 0.001 |
| Inpatient | 34.7 | 40.7 | 0.13[a] | 36.8 | 36.4 | 0.009 |
| Emergency | 45.8 | 46.9 | 0.02 | 45.9 | 45.7 | 0.004 |
| Virtual | 9.3 | 8.8 | 0.02 | 9.7 | 9.3 | 0.01 |

Shown were cohorts before and after propensity-score matching for the listed variables with their status based on the presence of related clinical codes anytime to the day of the index event. Shown were cohorts before and after propensity-score matching for the listed variables with their status based on the presence of related clinical codes anytime on or before the index event (the first prescription of semaglutide, or non-GLP-1RA anti-obesity medications during 6/2021–12/2022). Adverse socioeconomic determinants of health include problems related to education and literacy, employment and unemployment, housing and economic circumstances, social environment, upbringing, primary support group including family circumstances, certain psychosocial circumstances, and other psychosocial circumstances. Problems with lifestyle included tobacco use, lack of physical exercise, inappropriate diet and eating habits, high-risk sexual behavior, gambling and betting, and other problems related to lifestyle including antisocial behavior and sleep deprivation.
*SMD* standardized mean differences, *SD* standard deviation.
[a]SMD greater than 0.1, a threshold indicating cohort imbalance.

The underlying mechanisms have not been fully delineated but are likely to involve modulation of the brain dopamine reward system via GLP-1 receptors, which are present both in the ventral tegmental areas (VTA), where dopamine neurons are located, and in the nucleus accumbens (NAc), which is the main projection of VTA dopamine neurons[14]. The involvement of the dopamine reward pathway in modulating food and alcohol consumption[15] could explain why semaglutide is beneficial in reducing food consumption[16] and in animal models reducing alcohol and other drug consumption[5]. Indeed, semaglutide binds to the NAc[7] where it has been shown to attenuate alcohol-induced dopamine increases in alcohol drinking rats[7] providing evidence of semaglutide's modulation of the mesolimbic dopamine reward system[17]. Importantly the rewarding effects of food are a main contributor to overeating and obesity[18] just as the rewarding effects of alcohol drive alcohol consumption[19].

Because GLP-1 also mediates stress responses[20], this could be another mechanism by which semaglutide could buffer stress-related overeating and alcohol consumption[21]. The habenula, which has a high concentration of GLP1 receptors[22] could also participate in semaglutide's actions as it is involved in the negative reinforcement in obesity[23] and in alcohol and other substance use disorders[24]. Additionally, the anti-inflammatory effects of semaglutide and other GLP1-RA medication have also been implicated in its potential beneficial effects for AUD and other substance use disorder[6]. However, the beneficial effects of semaglutide for alcohol consumption could also reflect the fact that alcohol like food serves as a source of energy[25], and could include a combination of central[7] and peripheral mechanisms such as the effects of semaglutide on alcohol absorption, pharmacokinetics and metabolism[10]. Though there are no reports on semaglutide's effects on alcohol absorption and pharmacokinetics it is likely that since it decreases gastric emptying it would also likely decrease alcohol's absorption. Because the rate of alcohol absorption influences its rewarding effects[26], delayed absorption could make alcohol less rewarding. Delayed absorption could also increase alcohol's

(a)

**Incident AUD diagnosis in patients with obesity and no prior history of AUD**
**during 12–month follow–up time period**
**(comparison between propensity–score matched cohorts)**

| Population | semaglutide cohort | non–GLP–1RA anti–obesity medications cohort | | HR (95% CI) |
|---|---|---|---|---|
| Overall (n = 26,566/cohort) | 0.37% (98) | 0.73% (193) | | 0.50 (0.39–0.63) |
| Women (n = 17,977/cohort) | 0.22% (40) | 0.44% (79) | | 0.50 (0.34–0.73) |
| Men (n = 6,903/cohort) | 0.59% (41) | 1.14% (79) | | 0.50 (0.35–0.74) |
| age <= 55 years (n = 15,767/cohort) | 0.30% (48) | 0.61% (96) | | 0.49 (0.35–0.70) |
| age > 55 years (n = 10,440/cohort) | 0.48% (50) | 0.86% (90) | | 0.54 (0.38–0.76) |
| Black (n = 4,107/cohort) | 0.32% (13) | 0.71% (29) | | 0.43 (0.23–0.83) |
| White (n = 17,861/cohort) | 0.35% (62) | 0.67% (120) | | 0.51 (0.38–0.69) |
| No T2DM (n = 17,609/cohort) | 0.39% (68) | 0.60% (106) | | 0.64 (0.47–0.87) |
| T2DM (n = 8,696/cohort) | 0.30% (26) | 0.90% (78) | | 0.32 (0.20–0.49) |

0.10 0.20 0.40 0.80 2.0 4.0 8.00
Hazard Ratio (HR)

(b)

**Incident AUD diagnosis in patients with obesity and no prior history of AUD**
**during 12–month follow–up time period**
**(comparison between propensity–score matched cohorts)**

| Population | semaglutide cohort | naltrexone/topiramate cohort | | HR (95% CI) |
|---|---|---|---|---|
| Overall (n = 15,097/cohort) | 0.35% (53) | 0.78% (118) | | 0.44 (0.32–0.61) |
| Women (n = 10,718/cohort) | 0.24% (26) | 0.61% (65) | | 0.39 (0.25–0.62) |
| Men (n = 3,315/cohort) | 0.57% (19) | 1.36% (45) | | 0.41 (0.24–0.70) |
| age <= 55 years (n = 9,642/cohort) | 0.32% (31) | 0.72% (69) | | 0.44 (0.29–0.68) |
| age > 55 years (n = 5,289/cohort) | 0.45% (24) | 0.83% (44) | | 0.53 (0.32–0.87) |
| Black (n = 2,511/cohort) | <0.40% (<10) | 0.80% (20) | | 0.24 (0.09–0.65) |
| White (n = 9,808/cohort) | 0.38% (37) | 0.79% (77) | | 0.48 (0.32–0.71) |
| No T2DM (n = 11,335/cohort) | 0.40% (45) | 0.72% (81) | | 0.56 (0.39–0.80) |
| T2DM (n = 3,610/cohort) | <0.28% (<10) | 1.00% (36) | | 0.26 (0.13–0.53) |

0.10 0.20 0.40 0.80 2.0 4.0 8.00
Hazard Ratio (HR)

**Fig. 1 | Risk of incident AUD diagnosis in patients with obesity who had no prior history of AUD. a** Comparison between propensity-score matched semaglutide and non-GLP-1RA anti-obesity medications cohorts, stratified by gender, age group, race, and diagnosis of T2DM. **b** Comparison between propensity-score matched semaglutide and naltrexone/topiramate cohorts, stratified by gender, age group, race, and the diagnosis of T2DM. Patients were followed for 12 months after the index event (first prescription of semaglutide, non-GLP-1 RA anti-obesity medications, or naltrexone/topiramate during 6/2021–12/2022). Hazard rates were calculated using Cox proportional hazards analysis to estimate hazard rates of outcome at daily time intervals with censoring applied. Overall risk = number of patients with outcomes during the 12-month time window/number of patients in the cohort at the beginning of the time window. AUD Alcohol use disorders, GLP-1RA glucagon-like peptide-1 receptor agonist, T2DM type 2 diabetes. Source data are provided as a Source Data file.

metabolism in the stomach into acetaldehyde[27], which would enhance its aversive effects.

As of now, only one randomized clinical trial has been published that evaluated the effects of a GLP-1RA exenatide in patients with AUD[12]. Though this trial did not report reductions in heavy alcohol drinking days (main outcome), it showed a significant attenuation of brain activation to alcohol cues. Also, in a secondary analysis the investigators found a significant reduction in heavy drinking days and total alcohol intake in AUD patients with obesity. This is relevant to our findings since the benefits of semaglutide were observed in

**Table 2 | Characteristics of the semaglutide cohort and the anti-obesity medications cohort for the study population with obesity who had a prior history of AUD**

| | Before propensity-score matching | | | After propensity-score matching | | |
|---|---|---|---|---|---|---|
| | Semaglutide cohort | Non-GLP-1RA anti-obesity medications cohort | SMD | Semaglutide cohort | Non-GLP-1RA anti-obesity medications cohort | SMD |
| Total number | 1,470 | 2,784 | | 1,051 | 1,051 | |
| Age at index event (years, mean ± SD) | 53.6 ± 12.7 | 50.5 ± 12.9 | 0.25[a] | 52.7 ± 12.8 | 52.4 ± 13.0 | 0.02 |
| Sex (%) | | | | | | |
| Female | 41.4 | 36.1 | 0.11[a] | 40.9 | 42.0 | 0.02 |
| Male | 54.8 | 59.2 | 0.09 | 55.5 | 54.5 | 0.02 |
| Unknown | 3.7 | 4.7 | 0.05 | 3.6 | 3.5 | 0.005 |
| Ethnicity (%) | | | | | | |
| Hispanic/Latinx | 8.5 | 6.1 | 0.09 | 7.4 | 7.4 | <.001 |
| Not Hispanic/Latinx | 73.9 | 75.4 | 0.03 | 74.2 | 73.9 | 0.007 |
| Unknown | 17.6 | 18.5 | 0.02 | 18.4 | 18.4 | 0.007 |
| Race (%) | | | | | | |
| Asian | 1.8 | 0.7 | 0.09 | 1.4 | 1.0 | 0.04 |
| Black | 15.2 | 17.1 | 0.05 | 16.5 | 16.6 | 0.003 |
| White | 67.2 | 65.5 | 0.04 | 66.1 | 66.3 | 0.004 |
| Unknown | 10.9 | 11.5 | 0.02 | 11.6 | 11.5 | 0.003 |
| Marital status (%) | | | | | | |
| Never Married | 13.9 | 19.5 | 0.15[a] | 14.8 | 15.3 | 0.01 |
| Divorced | 6.9 | 6.1 | 0.03 | 6.6 | 6.9 | 0.01 |
| Widowed | 2.0 | 2.3 | 0.02 | 2.3 | 2.6 | 0.02 |
| Adverse socioeconomic determinants of health (%) | 14.9 | 18.9 | 0.11[a] | 14.8 | 15.0 | 0.005 |
| Problems related to lifestyle (%) | 28.2 | 32.1 | 0.09 | 30.5 | 29.2 | 0.03 |
| Obesity categories (%) | | | | | | |
| Morbid (severe) obesity due to excess calories | 63.6 | 38.3 | 0.52[a] | 56.1 | 56.9 | 0.02 |
| Obesity, unspecified | 80.3 | 73.1 | 0.17[a] | 77.6 | 77.7 | 0.002 |
| Other obesity due to excess calories | 19.0 | 11.5 | 0.21[a] | 16.7 | 16.5 | 0.005 |
| BMI 30.0–30.9 | 7.6 | 12.0 | 0.15[a] | 9.0 | 9.0 | <.001 |
| BMI 31.0–31.9 | 8.6 | 11.9 | 0.11[a] | 9.6 | 9.7 | 0.006 |
| BMI 32.0–32.9 | 10.8 | 12.5 | 0.05 | 11.1 | 12.2 | 0.03 |
| BMI 33.0–33.9 | 11.2 | 12.2 | 0.03 | 11.8 | 11.9 | 0.003 |
| BMI 34.0–34.9 | 13.1 | 12.9 | 0.005 | 12.9 | 13.4 | 0.01 |
| BMI 35.0–35.9 | 14.0 | 11.2 | 0.09 | 13.2 | 13.9 | 0.02 |
| BMI 36.0–36.9 | 14.8 | 10.1 | 0.14[a] | 13.6 | 13.7 | 0.003 |
| BMI 37.0–37.9 | 15.0 | 9.2 | 0.18[a] | 12.9 | 13.4 | 0.01 |
| BMI 38.0–38.9 | 13.8 | 7.7 | 0.20[a] | 11.2 | 11.6 | 0.01 |
| BMI 39.0–39.9 | 12.0 | 7.1 | 0.17[a] | 9.4 | 9.9 | 0.02 |
| BMI 40.0–44.9 | 25.5 | 16.8 | 0.21[a] | 22.9 | 23.1 | 0.005 |
| BMI 45.0–49.9 | 14.8 | 8.1 | 0.21[a] | 11.6 | 11.4 | 0.006 |
| BMI 50.0–59.9 | 9.2 | 5.1 | 0.16[a] | 7.2 | 7.5 | 0.01 |
| BMI 60.0–69.9 | 2.4 | 1.3 | 0.08 | 1.7 | 2.1 | 0.03 |
| BMI ≥70 | 1.4 | 0.6 | 0.07 | 1.2 | 1.1 | 0.009 |
| Family history of mental and behavioral disorders | 2.7 | 4.3 | 0.09 | 2.0 | 2.9 | 0.06 |
| Pre-existing medical conditions, procedures, medications (%) | | | | | | |
| Type 2 diabetes | 59.2 | 27.6 | 0.67[a] | 47.9 | 47.8 | 0.002 |
| Depression | 56.7 | 57.0 | 0.007 | 57.8 | 58.1 | 0.01 |
| Mood disorders | 64.1 | 68.2 | 0.09 | 65.6 | 67.0 | 0.03 |
| Anxiety disorders | 65.0 | 66.8 | 0.04 | 65.4 | 66.7 | 0.03 |
| Psychotic disorders | 6.4 | 9.9 | 0.13[a] | 7.2 | 7.4 | 0.007 |
| Behavioral disorders | 17.3 | 12.1 | 0.15[a] | 16.2 | 16.6 | 0.01 |
| Disorders of adult personality and behavior | 7.3 | 6.8 | 0.02 | 7.3 | 7.3 | <.001 |
| | 10.1 | 8.3 | 0.06 | 9.0 | 10.3 | 0.04 |

**Table 2 (continued) | Characteristics of the semaglutide cohort and the anti-obesity medications cohort for the study population with obesity who had a prior history of AUD**

| | Before propensity-score matching | | | After propensity-score matching | | |
|---|---|---|---|---|---|---|
| | Semaglutide cohort | Non-GLP-1RA anti-obesity medications cohort | SMD | Semaglutide cohort | Non-GLP-1RA anti-obesity medications cohort | SMD |
| Behavioral and emotional disorders with onset usually occurring in childhood and adolescence | | | | | | |
| Conduct disorders | 1.6 | 1.1 | 0.04 | 1.5 | 1.5 | <.001 |
| Symptoms and signs involving emotional state | 17.8 | 23.5 | 0.14[a] | 18.2 | 19.7 | 0.04 |
| Chronic pain | 47.3 | 35.7 | 0.24[a] | 43.1 | 44.7 | 0.03 |
| Cancer | 51.0 | 30.7 | 0.42[a] | 46.2 | 45.9 | 0.008 |
| Cannabis use disorder | 11.4 | 16.7 | 0.15[a] | 11.9 | 11.1 | 0.02 |
| Opioid use disorder | 10.5 | 11.5 | 0.03 | 11.7 | 11.0 | 0.02 |
| Tobacco use disorder | 42.9 | 53.5 | 0.21[a] | 45.1 | 45.7 | 0.01 |
| Cocaine use disorder | 7.3 | 11.1 | 0.13[a] | 8.1 | 7.5 | 0.02 |
| Other stimulant disorders | 4.6 | 6.9 | 0.10[a] | 4.7 | 4.5 | 0.009 |
| Other psychoactive substance related disorders | 13.6 | 17.5 | 0.11[a] | 14.7 | 15.0 | 0.01 |
| Hypertension | 85.0 | 82.0 | 0.08 | 83.2 | 82.6 | 0.02 |
| Disorders of lipoprotein metabolism and other lipidemias | 78.1 | 56.6 | 0.47[a] | 72.4 | 72.5 | 0.002 |
| Hyperlipidemia | 65.3 | 45.2 | 0.41[a] | 59.1 | 59.6 | 0.001 |
| Hypercholesterolemia | 28.9 | 16.1 | 0.31[a] | 23.9 | 24.7 | 0.02 |
| Ischemic heart diseases | 29.2 | 22.4 | 0.16[a] | 26.2 | 24.4 | 0.04 |
| Other forms of heart disease | 51.8 | 44.4 | 0.15[a] | 49.2 | 48.3 | 0.02 |
| Cerebral infarction | 5.9 | 5.1 | 0.03 | 5.6 | 4.8 | 0.04 |
| Cerebrovascular diseases | 12.4 | 11.2 | 0.04 | 11.7 | 12.2 | 0.02 |
| Substance abuse treatment | 2.7 | 8.0 | 0.24[a] | 3.4 | 3.2 | 0.01 |
| Psychotherapy | 16.8 | 11.4 | 0.16[a] | 14.7 | 15.8 | 0.03 |
| Acamprosate | 2.0 | 3.2 | 0.08 | 2.1 | 2.7 | 0.04 |
| Disulfiram | 2.5 | 1.9 | 0.04 | 2.2 | 2.1 | 0.007 |
| Baclonfen | 7.6 | 5.3 | 0.09 | 7.0 | 6.8 | 0.01 |
| Gabapentin | 40.5 | 36.5 | 0.08 | 39.5 | 39.4 | 0.002 |
| Medical visit types (%) | | | | | | |
| Outpatient | 97.1 | 90.4 | 0.28[a] | 96.3 | 96.7 | 0.02 |
| Inpatient | 59.5 | 65.5 | 0.13[a] | 59.4 | 58.6 | 0.02 |
| Emergency | 69.0 | 69.9 | 0.05 | 68.1 | 69.5 | 0.03 |
| Virtual | 12.9 | 12.1 | 0.03 | 12.4 | 13.8 | 0.04 |

Shown were cohorts before and after propensity-score matching for the listed variables with their status based on the presence of related clinical codes anytime to the day of the index event. Shown were cohorts before and after propensity-score matching for the listed variables with their status based on the presence of related clinical codes anytime on or before the index event (the first prescription of semaglutide, or non-GLP-1RA anti-obesity medications during 6/2021–12/2022). Adverse socioeconomic determinants of health include problems related to education and literacy, employment and unemployment, housing and economic circumstances, social environment, upbringing, primary support group including family circumstances, certain psychosocial circumstances, and other psychosocial circumstances. Problems with lifestyle included tobacco use, lack of physical exercise, inappropriate diet and eating habits, high-risk sexual behavior, gambling and betting, and other problems related to lifestyle including antisocial behavior and sleep deprivation.
*SMD* standardized mean differences, *SD* standard deviation.
[a]SMD greater than 0.1, a threshold indicating cohort imbalance.

patients with obesity and in patients with T2DM many of whom also had obesity. In the analysis of patients with T2DM stratified by their having or not having a diagnosis of obesity, we observed that the lower risk of incident AUD with semaglutide in patients without obesity was similar in patients with obesity. In summary, our study provides real-world evidence supporting the therapeutic benefits of semaglutide for AUD. It is important to clarify that our findings of lower risk of AUD incidence and relapse in patients taking semaglutide cannot be interpreted to indicate that semaglutide reduced AUD symptomatology and are insufficient to justify clinicians' use of semaglutide off-label to treat AUD. For this to happen data from randomized clinical trials are necessary. Currently, there are five registered clinical trials to evaluate the effect of semaglutide in AUD, and some are already recruiting[28–32]. Since individuals with AUD are at higher risk for mood disorders and suicidality[33,34] and there have

been concerns that semaglutide could increase these[35], though recent evidence suggests it decreases them[36], it will be important for future clinical trials to assess semaglutide's effects in mood and suicidal ideation. Future studies should also evaluate interactions with alcohol and with medications for AUD.

Our study has several limitations: First, this is a retrospective observational study, so no causal inferences can be drawn. Second, our study populations represented those who had medical encounters with healthcare systems contributing to the TriNetX Platform. Finding from this study need to be validated in other populations. Third, there are limitations inherent in retrospective observational studies including unmeasured or uncontrolled confounders, self-selection, reverse causality, and other biases. Although the findings were replicated in two separate study populations with different characteristics at two non-overlapping study periods and with non-overlapping exposure

(a)

**Recurrent AUD diagnosis in patients with obesity and a prior history of AUD during 12–month follow–up time period (comparison between propensity–score matched cohorts)**

| Population | semaglutide cohort | non–GLP–1RA anti–obesity medications cohort | | HR (95% CI) |
|---|---|---|---|---|
| Overall (n = 1,051/cohort) | 22.6% (238) | 43.0% (452) | | 0.44 (0.38–0.52) |
| Women (n = 420/cohort) | 19.0% (80) | 32.9% (138) | | 0.51 (0.39–0.67) |
| Men (n = 553/cohort) | 23.9% (132) | 46.5% (257) | | 0.42 (0.34–0.51) |
| age <= 55 years (n = 586/cohort) | 22.9% (134) | 43.9% (257) | | 0.44 (0.35–0.54) |
| age > 55 years (n = 440/cohort) | 23.2% (102) | 36.8% (162) | | 0.55 (0.43–0.70) |
| Black (n = 140/cohort) | 20.7% (29) | 37.1% (52) | | 0.49 (0.31–0.78) |
| White (n = 699/cohort) | 22.7% (159) | 41.5% (290) | | 0.46 (0.38–0.56) |
| No T2D (n = 540/cohort) | 20.6% (111) | 41.5% (224) | | 0.42 (0.33–0.52) |
| T2D (n = 453/cohort) | 24.3% (110) | 40.4% (183) | | 0.50 (0.39–0.63) |

0.10 0.20 0.40 0.80   2.0  4.0 8.00
Hazard Ratio (HR)

(b)

**Recurrent AUD diagnosis in patients with obesity and a prior history of AUD during 12–month follow–up time period (comparison between propensity–score matched cohorts)**

| Population | semaglutide cohort | naltrexone/topiramate cohort | | HR (95% CI) |
|---|---|---|---|---|
| Overall (n = 715/cohort) | 21.5% (154) | 59.9% (428) | | 0.25 (0.21–0.30) |
| Women (n = 291/cohort) | 17.5% (51) | 54.0% (157) | | 0.23 (0.17–0.32) |
| Men (n = 379/cohort) | 24.0% (91) | 66.2% (251) | | 0.23 (0.18–0.30) |
| age <= 55 years (n = 423/cohort) | 23.9% (101) | 62.6% (265) | | 0.26 (0.21–0.33) |
| age > 55 years (n = 260/cohort) | 23.1% (60) | 53.4% (144) | | 0.32 (0.23–0.43) |
| Black (n = 76/cohort) | 23.7% (18) | 51.3% (39) | | 0.38 (0.22–0.67) |
| White (n = 444/cohort) | 20.9% (93) | 58.6% (260) | | 0.25 (0.20–0.32) |
| No T2D (n = 282/cohort) | 20.2% (57) | 57.4% (162) | | 0.25 (0.19–0.34) |
| T2D (n = 134/cohort) | 25.4% (34) | 61.9% (83) | | 0.27 (0.18–0.41) |

0.10 0.20 0.40 0.80   2.0  4.0 8.00
Hazard Ratio (HR)

**Fig. 2 | Risk of recurrent AUD diagnosis in patients with obesity who had a prior history of AUD. a** Comparison between propensity-score matched semaglutide and non-GLP-1RA anti-obesity medications cohorts, stratified by gender, age group, race, and the status of T2DM. **b** Comparison between propensity-score matched semaglutide and naltrexone/topiramate cohorts, stratified by gender, age group, race, and diagnosis of T2DM. Patients were followed for 12 months after the index event (first prescription of semaglutide, non-GLP-1 RA anti-obesity medications, or naltrexone/topiramate during 6/2021–12/2022). Hazard rates were calculated using Cox proportional hazards analysis to estimate hazard rates of outcome at daily time intervals with censoring applied. Overall risk = number of patients with outcomes during the 12-month time window/number of patients in the cohort at the beginning of the time window. AUD Alcohol use disorders, GLP-1RA glucagon-like peptide-1 receptor agonist, T2DM type 2 diabetes. Source data are provided as a Source Data file.

cohorts, potential biases or confounders could not be fully eliminated in this observational study. Fourth the follow-up time for the main analyses was 12 months. For the study population with T2DM we conducted a longer follow-up - up to 3 years and observed consistently lower risks in both incident and recurrent AUD associated with semaglutide. However, future studies are necessary to evaluate longer-term associations of semaglutide with AUD in patients with obesity. Fifth, the weekly higher dose format of 2.4 mg semaglutide (marketed as Wegovy) was approved for weight management, and the lower dose format of 0.5–1 mg semaglutide (marketed as Ozempic) was approved

(a)

**Incident AUD diagnosis in patients with T2DM and no prior history of AUD**
**during 12–month follow–up time period**
**(comparison between propensity–score matched cohorts)**

| Population | semaglutide cohort | non–GLP–1RA anti–diabetes medications cohort | | HR (95% CI) |
|---|---|---|---|---|
| Overall (n = 25,670/cohort) | 0.32% (81) | 0.52% (134) | | 0.56 (0.43–0.74) |
| Women (n = 11,743/cohort) | 0.19% (22) | 0.34% (40) | | 0.52 (0.31–0.88) |
| Men (n = 11,833/cohort) | 0.41% (49) | 0.73% (86) | | 0.53 (0.38–0.76) |
| age <= 55 years (n = 9,974/cohort) | 0.34% (34) | 0.53% (53) | | 0.60 (0.39–0.93) |
| age > 55 years (n = 15,951/cohort) | 0.30% (47) | 0.53% (84) | | 0.53 (0.37–0.76) |
| Black (n = 3,752/cohort) | 0.35% (13) | 0.51% (19) | | 0.64 (0.31–1.29) |
| White (n = 15,452/cohort) | 0.28% (43) | 0.58% (90) | | 0.45 (0.31–0.65) |
| No obesity (n = 10,112/cohort) | 0.33% (33) | 0.58% (59) | | 0.51 (0.33–0.78) |
| Obesity (n = 15,551/cohort) | 0.31% (48) | 0.47% (73) | | 0.63 (0.44–0.90) |

0.10 0.20 0.40 0.80  2.0  4.0 8.00
Hazard Ratio (HR)

(b)

**Recurrent AUD diagnosis in patients with T2DM and a prior history of AUD**
**during 12–month follow–up time period**
**(comparison between propensity–score matched cohorts)**

| Population | semaglutide cohort | non–GLP–1RA anti–diabetes medications cohort | | HR (95% CI) |
|---|---|---|---|---|
| Overall (n = 653/cohort) | 23.4% (153) | 33.2% (217) | | 0.61 (0.50–0.75) |
| Women (n = 163/cohort) | 20.9% (34) | 26.4% (43) | | 0.73 (0.47–1.15) |
| Men (n = 443/cohort) | 22.1% (98) | 31.2% (138) | | 0.61 (0.47–0.79) |
| age <= 55 years (n = 258/cohort) | 26.7% (69) | 37.6% (97) | | 0.61 (0.45–0.84) |
| age > 55 years (n = 402/cohort) | 20.6% (83) | 32.8% (132) | | 0.55 (0.42–0.72) |
| Black (n = 104/cohort) | 29.8% (31) | 34.6% (36) | | 0.78 (0.49–1.27) |
| White (n = 370/cohort) | 19.7% (73) | 29.7% (110) | | 0.58 (0.43–0.78) |
| No obesity (n = 195/cohort) | 26.2% (51) | 41.5% (81) | | 0.52 (0.37–0.74) |
| Obesity (n = 463/cohort) | 21.2% (98) | 33.3% (154) | | 0.55 (0.43–0.71) |

0.10 0.20 0.40 0.80  2.0  4.0 8.00
Hazard Ratio (HR)

(c)

**Incident and recurrent AUD diagnosis in patients with T2DM**
**at longer follow–up time period**
**(comparison between propensity–score matched cohorts)**

| Follow–up | semaglutide cohort | non–GLP–1RA anti–diabetes medications cohort | | HR (95% CI) |
|---|---|---|---|---|
| **Incident AUD (n=25,670/cohort)** | | | | |
| 1–year | 0.32% (81) | 0.52% (134) | | 0.56 (0.43–0.74) |
| 2–year | 0.58% (149) | 0.90% (231) | | 0.59 (0.48–0.73) |
| 3–year | 0.90% (232) | 1.19% (305) | | 0.72 (0.60–0.85) |
| **Recurrent AUD (n=653/cohort)** | | | | |
| 1–year | 23.4% (153) | 33.2% (217) | | 0.61 (0.50–0.75) |
| 2–year | 29.3% (191) | 38.9% (294) | | 0.64 (0.53–0.77) |
| 3–year | 33.5% (219) | 41.7% (272) | | 0.68 (0.57–0.81) |

0.10  0.30 0.80 2.0  5.00
Hazard Ratio (HR)

for treating T2DM). Interestingly we observed a stronger association of semaglutide with recurrent AUD in patients with obesity than in patients with T2DM (HR of 0.53 vs. 0.74), which could suggest a potential dosage effect. However, the characteristics of these 2 study populations, the comparators, and the study periods were different. Since different dose forms of semaglutide were approved for different

disease indications, we could not directly examine the dosage effect of semaglutide in our study.

In summary, our results find an association between reduced risk for incident and AUD relapse with the prescription of smaglutide in patients with obesity or T2DM. While these findings provide preliminary evidence of the potential benefit of semaglutide in AUD in

**Fig. 3 | Risk of incident and recurrent AUD diagnosis in patients with T2DM.**
**a** Comparison of 12-month risk for incident AUD diagnosis between propensity-score matched semaglutide and non-GLP-1RA anti-diabetes medications cohorts, stratified by gender, age group, race, and the diagnosis of obesity. **b** Comparison of 12-month risk of recurrent AUD diagnosis between propensity-score matched semaglutide and non-GLP-1RA anti-diabetes medications cohorts, stratified by gender, age group, race, and the diagnosis of obesity. **c** Comparison of longer-term risks of incident and recurrent AUD diagnosis between propensity-score matched semaglutide and non-GLP-1RA anti-diabetes medications cohorts. Patients were followed for 12 months, 2-year and 3-year after the index event (first prescription of semaglutide, non-GLP-1 RA anti-diabetes medications in 12/2017–5/2021). AUD Alcohol use disorders, GLP-1RA glucagon-like peptide-1 receptor agonist, T2DM type 2 diabetes. Source data are provided as a Source Data file.

real-world populations further randomized clinical trials are needed to support its use clinically for AUD.

## Methods
### Database
We used built-in statistical and informatics functions within the TriNetX Analytics Platform[37] (Research US Collaborative Network) to analyze aggregated and de-identified patient electronic health records (EHRs). Analyses were performed on January 26, 2024. At the time of this study, TriNetX Research US Collaborative Network contained EHRs of 105.3 million patients from 61 healthcare organizations, most of which are large academic medical institutions, in the US across 50 states: 25%, 17%, 41%, and 12% in the Northeast, Midwest, South, West, respectively, and 5% unknown region. We previously used the TriNetX platform to perform retrospective cohort studies[36,38–51] in various populations including patients with substance use disorders[38,45,46,48,51]. We also used the TriNetX platform to examine the associations of GLP-1RAs with colorectal cancer[50] and semaglutide with suicidal ideations[36] and cannabis use disorder[51].

TriNetX de-identifies and aggregates EHRs from contributing healthcare systems completes an intensive data preprocessing stage to minimize missing values, maps the data to a common clinical data model, and provides web-based analytics tools to analyze patient EHRs. All variables are either binary, categorical, or continuous but essentially guaranteed to exist. Missing sex, race, and ethnicity values are represented using "Unknown Sex", "Unknown race" and "Unknown Ethnicity", respectively. For other variables (e.g., medical conditions, medications, procedures, lab tests, and socio-economic determinant health), the value is either present or absent, and "missing" is not pertinent.

### Ethics statement
The TriNetX platform aggregates and HIPAA de-identifies data contributed from the electronic health records of participating healthcare organizations. The TriNetX platform also only reports population-level results (no access to individual patient data) and uses statistical "blurring", reporting all population-level counts between 1 and 10 as 10. Based on the de-identification methods used by TriNetX, as per HIPAA privacy and security rules[52], TriNetX sought and obtained expert attestation that TriNetX data is HIPAA de-identified. Because the data in the TriNetX platform is HIPAA de-identified, and therefore, "by definition" is deemed to allow no access to protected health information (and therefore no risk of protected health information disclosure), Institutional Review Boards (IRBs) have no jurisdiction of studies using HIPAA de-identified data[53]. Since the study concerns non-human subject research, consent from participants was waived and IRB approval was not required for this study.

### Study populations
**The study population with obesity.** The analyses for the association of semaglutide with both incident and recurrent diagnosis of AUD in patients with obesity were restricted to a starting date of 6/2021 when semaglutide was approved in the US for weight management as Wegovy and an ending date of 12/2022, which allowed for a 12-month follow-up period by the time of data collection and analysis on January 26, 2024.

To assess the associations of semaglutide with incident AUD (first time diagnosis of AUD), the study population included 83,825 patients who had active medical encounters for the diagnosis of obesity in 6/2021–12/2022, were for the first time (new-user design) prescribed semaglutide or non-GLP-1RA anti-obesity medications (naltrexone, topiramate, bupropion, orlistat, phentermine)[54] during 6/2021–12/2022 (time zero or index event), had no diagnosis of AUD on or before the index event and had a diagnosis of at least one of obesity-associated comorbidities (T2D, hypertension, hypercholesterolemia, hyperlipidemia, heart diseases, stroke) on or before the index event. Patients who were prescribed other GLP-1RAs or had bariatric surgery on or before the index event were excluded. This study population was then divided into 3 cohorts: (1) semaglutide cohort – 45,797 patients who were first-time prescribed semaglutide, (2) non-GLP1-RA anti-obesity medication cohort – 38,028 patients who were first-time prescribed non-GLP-1RA anti-obesity medications but not semaglutide and (3) naltrexone/topiramate cohort – 16,676 patients who were first time prescribed naltrexone and topiramate but not semaglutide. Among the non-GLP-1RA anti-obesity medications, naltrexone and topiramate were also prescribed for AUD[3]. We constructed the naltrexone/topiramate cohort to compare semaglutide to naltrexone/topiramate for incident AUD risk in patients with obesity. We used new-user design to mitigate prevalent user bias and confounding associated with the drug itself[55,56].

To assess the associations of semaglutide with recurrent AUD diagnosis (recurrent medical encounters for AUD diagnosis), the study population included 4254 patients who had active medical encounters for the diagnosis of obesity in 6/2021–12/2022, were for the first time (new-user design) prescribed semaglutide or non-GLP-1RA anti-obesity medications during 6/2021–12/2022 (index event), had a diagnosis of AUD on or before the index event and had a diagnosis of at least one of obesity-associated comorbidities on or before the index event. Patients who were prescribed other GLP-1RAs or had bariatric surgery on or before the index event were excluded. This study population was then divided into 3 cohorts: (1) semaglutide cohort – 1470 patients who were first-time prescribed semaglutide, (2) non-GLP1-RA anti-obesity medication cohort – 2784 patients who were first-time prescribed non-GLP-1RA anti-obesity medications but not semaglutide and (3) naltrexone/topiramate cohort – 1430 patients who were first time prescribed naltrexone and topiramate but not semaglutide. We constructed the naltrexone/topiramate cohort to compare semaglutide to naltrexone/topiramate for recurrent AUD risk in patients with obesity.

**The study populations with T2DM.** The analyses on the associations of semaglutide with both incident and recurrent AUD among patients with T2DM had a starting time of 12/2017 when semaglutide was approved in the US to treat T2DM as Ozempic and an ending date of 5/2021 to allow us to separately examine the associations of semaglutide on AUD as Ozempic from those as Wegovy in the study population with obesity. Since patients in the study population with obesity were for the first time prescribed semaglutide after 6/2021, there was no overlap in the exposure cohorts for these two study populations.

To assess the association of semaglutide with incident AUD, the study population included 598,803 patients with T2DM who had

active medical encounters for T2DM during 12/2017–5/2021, were for the first time prescribed semaglutide or non-GLP1-1RA anti-diabetes medications (new-user design) during 12/2017–5/2021(index event), had no diagnosis of AUD on or before the index event and had a diagnosis of at least one of obesity-associated comorbidities (hypertension, hypercholesterolemia, hyperlipidemia, heart diseases, stroke) on or before the index event. The status of non-GLP1RA anti-diabetes medications was determined by the Anatomical Therapeutic Chemical or ATC code A10 "Drugs used in diabetes" with GLP-1RAs (ATC code A10BJ "Glucagon-like peptide-1 (GLP-1) analogs") excluded. The list of non-GLP1RA anti-diabetes medications included insulins, metformin, sulfonylureas, alpha glucosidase inhibitors, thiazolidinediones, dipeptidyl peptidase 4 (DPP-4) inhibitors, sodium-glucose co-transporter 2 (SGLT2) inhibitors. Patients who were prescribed other GLP-1RAs or had bariatric surgery on or before the index event were excluded. This study population was divided into two cohorts: (1) semaglutide cohort – 25,686 patients prescribed semaglutide, and (2) Non-GLP-1RA anti-diabetes medication cohort – 573,117 patients prescribed non-GLP-1RA anti-diabetes medications.

To assess the associations of semaglutide with recurrent AUD, the study population comprised 22,113 patients who had active medical encounters for T2DM diagnosis in 12/2017–5/2021, were for the first time prescribed semaglutide or non-GLP1-1RA anti-diabetes medications during 12/2017–5/2021(index event), had a diagnosis of AUD on or before the index event, and had a diagnosis of at least one of obesity-associated comorbidities on or before the index event. Patients who were prescribed other GLP-1RAs or had bariatric surgery on or before the index event were excluded. This study population was then divided into two cohorts: (1) semaglutide cohort – 668 patients prescribed semaglutide, and (2) non-GLP1-RA anti-diabetes medication cohort – 21,445 patients prescribed non-GLP-1RA anti-diabetes medications.

**Statistical analysis.** For each study population, the semaglutide cohort and the comparision cohort were propensity-score matched (1:1 using nearest neighbor greedy matching with a caliper of 0.25 times the standard deviation) on covariates that are potential risk factors for AUD[57–60] including demographics, adverse socio-economic determinants of health (e.g., problems related to education and literacy, employment and unemployment, housing and economic circumstances, social environment, upbringing, primary support group including family circumstances and various psycho-social circumstances), problems with lifestyle (e.g., tobacco use, lack of physical exercise, inappropriate diet and eating habits, high-risk sexual behavior, gambling and betting, and other problems related to lifestyle including antisocial behaviors and sleep deprivation), pre-existing medical conditions, medications, medical procedures and medical visit types (outpatient, inpatient, emergency, and virtual). Obesity sub-categories were also matched to control obesity severity which included 3 ICD-10 diagnosis codes and 15 BMI categories ranging from BMI 30 to BMI 70 or greater.

The outcome –incident or recurrent diagnosis of AUD (International Classification of Diseases, Tenth Revision (ICD-10) code F10 "Alcohol related disorders") – that occurred within the 12-month time window after the index events were compared between matched semaglutide and comparison cohorts. Cox proportional hazards analysis was used to estimate hazard rates of outcome at daily time intervals with censoring applied. When the last fact (the outcome of interests or other medical encounters) in the patient's record is in the time window for analysis, the patient was censored on the day after the last fact in their record. Hazard ratio (HR) and 95% confidence intervals were used to describe the relative hazard of the outcomes based on a comparison of time to event rates.

Separate analyses were performed in patients stratified by sex (women, men), age groups (≤55, >55 years), and race (Black, White). For the study population with obesity, a separate analysis was performed in patients with T2DM and patients without T2DM. Given that the previous clinical trial of the GLP-1RA exenatide for AUD found reduced alcohol consumption only in those who were overweight[12], we further separately examined the association of semaglutide with both incident and recurrent AUD in patients with T2DM, with and without obesity.

To examine longer-term associations of semaglutide with AUD, the outcome –incident and recurrent diagnosis of AUD– in patients with T2DM was further followed for 2-year, 3-year starting after the index event.

The data were collected and analyzed on January 26, 2024 within the TriNetX Analytics Platform using built-in functions (propensity-score matching, Cox proportional hazard analysis, Kaplan-Meier survival) implemented using Survival 3.2-3 in R 4.0.2 and libraries/utilities for data science and statistics in Python 3.7 and Java 11.0.16. Details of clinical codes for eligibility criteria, exposure, outcomes, and confounders are in Supplementary Table 3.

### Reporting summary
Further information on research design is available in the Nature Portfolio Reporting Summary linked to this article.

## Data availability
This study used population-level aggregate and HIPAA de-identified data collected by the TriNetX platform and available from TriNetX, LLC (https://trinetx.com/), but third-party restrictions apply to the availability of these data. The data were used under license for this study with restrictions that do not allow for the data to be redistributed or made publicly available. To gain access to the data, a request can be made to TriNetX (join@trinetx.com), but costs may be incurred, and a data-sharing agreement may be necessary. Data specific to this study including diagnosis codes and cohort characteristics in aggregated format are included in the manuscript as tables, figures, and supplementary files. Data through the TriNetX platform is queried in real-time with results being returned typically in seconds to minutes. Data from the underlying electronic health records of participating healthcare organizations is refreshed in the TriNetX platform from daily to every couple of months depending on the healthcare organization. Source data are provided with this paper.

## Code availability
All the statistical analyses in this study including propensity-score matching, and Cox proportional hazards used web-based built-in functions within the TriNetX Analytics Platform that are implemented using Survival 3.2-3 in R 4.0.2 and libraries/utilities for data science and statistics in Python 3.7 and Java 11.0.16. Data and code to recreate figures in the study can be accessed at https://github.com/bill-pipi/semaglutide_AUD

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

## Acknowledgements

We acknowledge support from the National Institute on Alcohol Abuse and Alcoholism (AA029831), National Institute on Aging (AG057557, AG061388, AG062272, AG07664), from National Cancer Institute Case Comprehensive Cancer Center (CA221718, CA043703, CA2332216)

## Author contributions

R.X. conceived the study. R.X. and N.D.V. designed the study. R.X., N.D.V. and W.W. interpreted the results and drafted the manuscript. W.W. performed data analysis and created tables and figures. N.A.B., P.B.D., and D.C.K. critically contributed to study design, result interpretation, and manuscript preparation. We confirm the originality of the content. R.X. had full access to all the data in the study and take responsibility for the integrity of the data and the accuracy of the data analysis.

## Competing interests

The authors declare no competing interests.
