## [Peer Review File · Nature Communications]

Associations of semaglutide with incidence and recurrence of alcohol use disorder in real-world populationREVIEWER COMMENTS

Reviewer #1 (Remarks to the Author):

These data are interesting, and timely. I think the study was well executed and I only have minor comments:

- It would be interesting to add information on how common the comorbidity of AUD and type 2 diabetes or obesity is.
- The authors write "reports from patients prescribed semaglutide describe reduced desire to drink9.". To the best of my knowledge, this is a report by a journalist rather than scientific study. This must be stated better. This concern is also relevant for the discussion.
- As stated in discussion, some of the T2DM were overweight or are some lean, would be interesting to mention general information in this also in introduction.
- The authors discuss the potential role of NAc. One study has found that semaglutide binds to NAc. Perhaps this reference would fit in this part of the discussion.
- The authors discuss the potential role of peripheral mechanics. Perhaps the study, in which central rather than GLP-1 receptors were implied for alcohol responses, could be added in this part of the discussion?

I therefore recommend minor revision.

Reviewer #2 (Remarks to the Author):

Thank you for the well written manuscript

A few questions, which should be discussed in the text:

There is anecdotal evidence that semaglutide can precipitate psychological/psychiatric disease - what is the current status of this potential adverse effect? How does it impact the usage of semaglutide as an anti addictive medication?

Semaglutide slows gastric emptying. Does it slow transit, and thereby increase absorption, of alcohol?

Can semaglutide reduce craving for tobacco and other street drugs as well?

Minor points - please use semaglutide with small case s throughout the text. Please change the trade names to semaglutide 0.25-1.0 mg and 2.4 mg, as appropriate

Reviewer #3 (Remarks to the Author):

This study leverages an EHR platform to examine observational associations between GLP1 agonist exposure and both the development and recurrence of AUD in obese and T2DM populations, finding consistent support in both cases. The manuscript has many strengths and is forthright about its limitations. I believe the following issues should be addressed:

1. My biggest concern is about the validity of the EHR platform. Am I correct it contains health record for 1/3 of the US population? This seems implausible and more information about the nature and validity of the platform is needed. The authors cite previous reports using it but evidence of internal validity about the quality/accuracy of these diagnoses is warranted.
2. For incidence of AUD in the obesity sample, the control medications include efficacious medications for alcohol – naltrexone and topiramate – which should be removed for clearer signal or evaluated as active placebos (i.e., evidence that both also reduce incidence would be an internal validity check and would provide a comparative effect size for contrast with semaglutide).
3. After the obesity incidence sample, specific details about the control medications are not provided but should be. Again, if AUD medications are included, this is potentially problematical/confounding.
4. I know this is an EHR study but is any further information about the patients with active AUD who are then prescribed a GLP1 agonist. This is quite an unusual clinical population and further information would be valuable (e.g., inpatient vs outpatient, addiction tx patients)
5. Why did the obesity AUD sample go from 1662 to 1577 during PSM? Likewise, why did T2DM AUD cohort go from 704 to 698?
6. There are missing words here and there that should be corrected.

AUTHOR'S RESPONSE TO REVIEWERS

REVIEWER COMMENTS

Reviewer #1 (Remarks to the Author):

These data are interesting, and timely. I think the study was well executed and I only have minor comments:

- It would be interesting to add information on how common the comorbidity of AUD and type 2 diabetes or obesity is.

Response: The detailed information is now included in Tables 1-2 for the obesity population and in the Extended Tables 1-2 for the T2DM population.

- The authors write "reports from patients prescribed semaglutide describe reduced desire to drink⁹". To the best of my knowledge, this is a report by a journalist rather than scientific study. This must be stated better. This concern is also relevant for the discussion.

Response: We reworded as follows "Anecdotal reports from patients prescribed semaglutide describe reduced desire to drink⁹ that have been corroborated by two recent clinical reports, one reporting reduced alcohol drinking with semaglutide or tirzepatide based on analyses of social media texts and follow up of selected participants (Quddos et al., 2023), and the other reporting decreased symptoms of in a case series of AUD patients treated with semaglutide (Richards et al 2024)." and included the relevant references:

Please see page 3 lines 65-69, page 5 lines 177-181.

- As stated in discussion, some of the T2DM were overweight or are some lean, would be interesting to mention general information in this also in introduction.

Response: We now include in the introduction the percentage of the patients with T2DM who are lean versus those who are overweight and also of patients with overweight/obesity who have T2DM and those who were not.

Please see page 3 lines 78-82.

- The authors discuss the potential role of NAc. One study has found that semaglutide binds to NAc. Perhaps this reference would fit in this part of the discussion.

Response: We now include the reference that reports binding of semaglutide in the NAc (Aranas et al., 2023) in this part of the discussion.

Please see page 6 lines 195-196.

- The authors discuss the potential role of peripheral mechanics. Perhaps the study, in which central rather than GLP-1 receptors were implied for alcohol responses, could be added in this part of the discussion?

Response: We now include the reference that implicates central effects of semaglutide on reducing alcohol drinking (Aranas et al., 2023) in this part of the discussion.

Please see page 6 lines 209-210.

I therefore recommend minor revision.

Reviewer #2 (Remarks to the Author):

Thank you for the well written manuscript

A few questions, which should be discussed in the text:

There is anecdotal evidence that semaglutide can precipitate psychological/psychiatric disease - what is the current status of this potential adverse effect? How does it impact the usage of semaglutide as an anti addictive medication?

Response: There is anecdotal evidence that semaglutide can precipitate anxiety, depressive symptoms and suicidal thoughts and self-harm but also that it can improve them. Most notably in July 2023, the European Medicines agency received reports of suicidal thoughts and self-harm in people prescribed semaglutide. To address this concern we conducted cohort studies similar to the ones in this study and showed that semaglutide was associated with a 50%-80% reduction in both incident and recurrent suicidal thoughts and published the findings in Nature Medicine on Jan 5, 2024. Six days after the publication, the FDA cleared semaglutide of suicidal risk. In parallel clinical evidence has emerged that GLP-1RA drugs such as semaglutide decrease depressive symptoms raising questions about their potential benefits as antidepressants (Chen et al., 2024). Since individuals with substance use disorders are at higher risk for mood disorders and suicidality it will be important for future clinical trials testing semaglutide to monitor mood and to also investigate its interactions with addictive drugs and with the medications for substance use disorders.

Please see pages 6-7 lines 232-236.

Semaglutide slows gastric emptying. Does it slow transit, and thereby increase absorption, of alcohol?

Response: Semaglutide decreases gastric emptying. While there are no reports on semaglutide's effects on alcohol pharmacokinetics it is likely that it would also decrease alcohol emptying. However this is more likely to slow its absorption reducing its rewarding effects (de Wit et al., 1992) and increasing its metabolism in the stomach into acetaldehyde (Yin et al., 1997) enhancing aversive effects, than of increasing the blood alcohol levels. In fact the slowed alcohol emptying has been proposed as a mechanism underlying the reduced consumption of alcohol with semaglutide (Quddos et al., 2023).

Please see pages 6 lines 210-215.

Can semaglutide reduce craving for tobacco and other street drugs as well?

Response: We are currently investigating the effects of semaglutide on tobacco smoking, cannabis use disorders and other substance use disorders and have two manuscripts under review.

Minor points - please use semaglutide with small case s throughout the text. Please change the trade names to semaglutide 0.25-1.0 mg and 2.4 mg, as appropriate

Response: We edited semaglutide, so it is now reported in small cases and edited the tradenames to semaglutide 0.25-1.0 mg instead of Ozempic and semaglutide 2.4 mg instead of Wegovy.

Please see pages 7 lines 250-252.

Reviewer #3 (Remarks to the Author):

This study leverages an EHR platform to examine observational associations between GLP1 agonist exposure and both the development and recurrence of AUD in obese and T2DM populations, finding consistent support in both cases. The manuscript has many strengths and is forthright about its limitations. I believe the following issues should be addressed:

1. My biggest concern is about the validity of the EHR platform. Am I correct it contains health record for 1/3 of the US population? This seems implausible and more information about the nature and validity of the platform is needed. The authors cite previous reports using it but evidence of internal validity about the quality/accuracy of these diagnoses is warranted.

Response: The TriNetX EHR platform contains longitudinal EHRs of 100 million unique patients across 20 years. For every single year, the number of patients is fewer than 100 million. The following are the number of patients who had active medical encounters with healthcare organizations in the TriNetX Network by calendar year:

2017: 23.8 million

2018: 26.4 million

2019: 27.7 million

2020: 28.2 million

2021:32.5 million

2022:30.9 million

2023:25.6 million

2017-2023: 74.6 million

We also cited a 2023 article by TriNetX team that has detailed description of the platform.

2. For incidence of AUD in the obesity sample, the control medications include efficacious medications for alcohol – naltrexone and topiramate – which should be removed for clearer signal or evaluated as active placebos (i.e., evidence that both also reduce incidence would be an internal validity check and would provide a comparative effect size for contrast with semaglutide.

Response: For both incident and recurrent AUD in patients with obesity, we performed additional analyses to compare semaglutide to naltrexone/topiramate (Figure 1b, Figure 2b). For both analyses, semaglutide was associated with significantly lower risks for incidence and recurrent AUD compared with naltrexone/topiramate.

Please see page 21, lines 546-549 and lines 563-564
Page 3-4, lines 94-99, 105-108, 121-127, 132-136
And updated Figure 1b, Figure 2b.

3. After the obesity incidence sample, specific details about the control medications are not provided but should be. Again, if AUD medications are included, this is potentially problematical/confounding.

Response: For the obesity population, we used two control medications cohorts: (1) non-GLP-1 anti-obesity medications (naltrexone, topiramate, bupropion, orlistat, phentermine and (2) naltrexone/ topiramate. Semaglutide was separately compared to these two comparison cohorts. Due to limited sample sizes, semaglutide was not compared to each of these medications. More details are now provided in the Method section.

4. I know this is an EHR study but is any further information about the patients with active AUD who are then prescribed a GLP1 agonist. This is quite an unusual clinical population and further information would be valuable (e.g., inpatient vs outpatient, addiction tx patients)

Response: We now include detailed information (70 variables) on patients with obesity who had a prior diagnosis of AUD in Table 2. Including demographics, marriage status, adverse socioeconomic determinants of health, problems related to lifestyle, comorbidities, medications, procedures, addiction treatment, medical visit types (inpatient, outpatient, emergency and virtual).

Similarly, Extended Table 2 provided detailed information about patients with T2DM who had a prior diagnosis of AUD.

Please see updated Tables 2 and Extended Table 2

5. Why did the obesity AUD sample go from 1662 to 1577 during PSM? Likewise, why did T2DM AUD cohort go from 704 to 698?

Response: This is because not all patients in the cohorts could be matched.

For example, in the revised study, for the study population of patients with obesity who had a prior diagnosis of AUD, there were 1,470 in the semaglutide cohort and 2,784 in the comparison cohort. After matching, 1,051 out of the 1,470 patients from the semaglutide cohort were matched to 1,051 out of the 2,784 patients in the comparison cohort.

6. There are missing words here and there that should be corrected.
Response: We corrected the grammatical errors.

REVIEWERS' COMMENTS:

Reviewer #1 (Remarks to the Author):

The authors have addressed my previous comments and the article is ready for publication.

Reviewer #2 (Remarks to the Author):

thank you

Reviewer #3 (Remarks to the Author):

The authors have capably addressed the points I raised. One minor note is that the final figure states CUD rather than AUD.